# A Study on Sample Diversity in Generative Models: GANs vs. Diffusion Models

**Reza Bayat**
Mila, Université de Montréal
`reza.bayat@mila.quebec`

## Abstract

In this project, we compare the sample diversity of two generative models: Generative Adversarial Networks (GANs) and Denoising Diffusion Probabilistic Models (DDPMs). GANs have achieved impressive results in generating high-quality samples, but have been known to suffer from the issue of mode collapse, which can result in a lack of sample diversity. Mode collapse occurs when the generator network in a GAN becomes stuck in a local minimum, causing it to produce samples that are similar to each other rather than sampling from the full range of possibilities in the target distribution. This can lead to a lack of sample diversity, as the generator is unable to explore and represent the full range of features in the data. DDPMs, on the other hand, have demonstrated improved sample diversity compared to GANs. We conducted experiments using both synthetic and image data to explore the connection between mode collapse and sample diversity in these two frameworks. Our findings indicate that by addressing the mode collapse problem, DDPM preserves a comprehensive representation of the distribution.

## 1 Introduction

Generative models are a type of machine learning model that is trained to learn and represent complex, high-dimensional probability distributions using data. Generative models can be used to estimate the probability of a particular observation and to generate new samples that follow the same underlying distribution as the training data Ruthotto & Haber (2021). Two of the most popular approaches of Generative models are Generative Adversarial Networks (GAN) Goodfellow et al. (2014) and Denoising Diffusion Probability Models (DDPM) Ho et al. (2020), which are the subjects of our study in this paper.

Sample diversity is an important property of generative models Metz et al. (2016), as it determines the range of possibilities that the model is able to capture and represent. A model with high sample diversity is able to generate a wide range of samples that capture the full range of features in the data, while a model with low sample diversity is limited to generating a narrow range of samples. Therefore, the objective of this study is to compare the sample diversity of Vanilla GAN and DDPM and investigate the correlation between mode collapse and sample diversity in these models.

## 2 Background and Settings

Given that we assume the reader of this study possesses knowledge of GANs and DDPMs, we have refrained from providing extensive explanations in this section due to space constraints. Nonetheless, we have included sections discussing these models and mode collapse in Appendix C.

## 3 Experiments

### 3.1 Synthetic dataset

We start with a simple dataset which is once a mixture of $8$ Gaussians with the means symmetrically arranged over a circle with a radius of $2$ and a standard deviation of $0.02$ and once more with $2$ disproportionately sampled Gaussians. Figure 1 shows the results of the DDPM and GAN models. While GAN struggles to generate samples from all modes during different iterations of training, DDPM can cover the whole space and generate adequate examples of each of them. In Figure 4 of Appendix A.2 it is visible that GAN collapses into the mode with more samples, DDPM rela-

tively succeeds to keep the balance between the generated samples. To find additional experiments conducted on different datasets, such as one Gaussian with a diagonal shape and grid of Gaussians, please look at Figure 2 (**Right**). More details of sysnthetic datasets can be found in Appendix A.1.

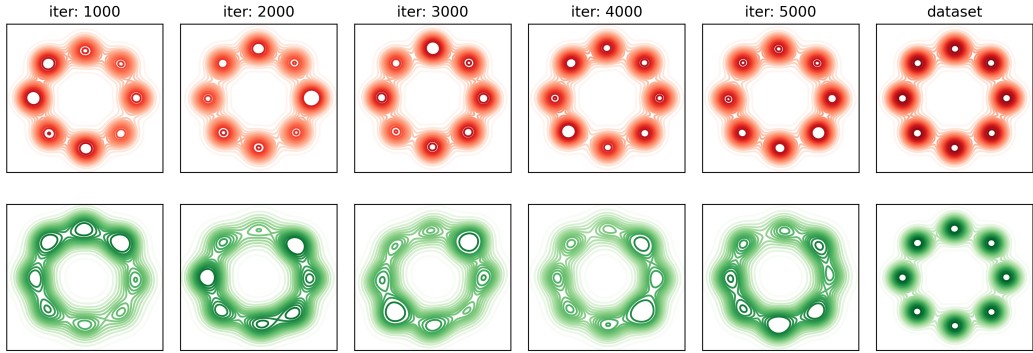

Figure 1: Samples of GAN (Green) and DDPM (Red) during the training on 8 Gaussians.

## 3.2 MNIST DATASET

We also ran a comparison of two models over a more complex dataset (MNIST), including hand-written digits along with their labels. As the results in Figure 2 (**Left**) show although GAN has not collapsed entirely on a single cluster of data it yet fails to maintain diversity of samples across the classes of data. On the other hand, DDPM keeps the variety of between the generated samples.

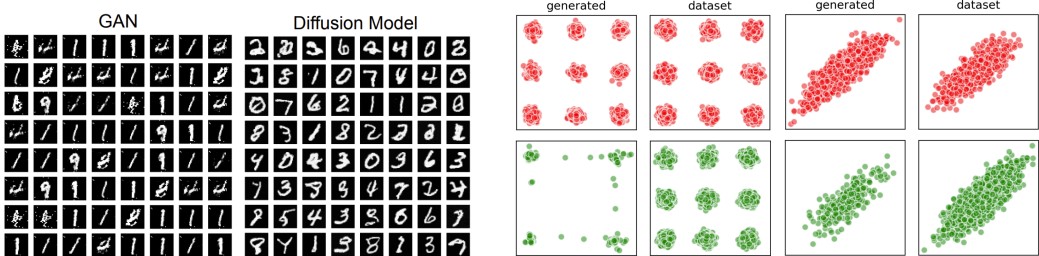

Figure 2: **Left:** Samples for MNIST dataset. Comparing to GAN, DDPM samples are more diverse. **Right:** Samples of GAN (Green) and DDPM of grid of Guassians and diagonally directed Gaussian.

In conclusion, the behavior of a *reverse* diffusion process is demonstrated in Figure 3 by two Gaussians that are of different sizes, which show the ability of DDPMs for covering the whole distribution as the time step progress.

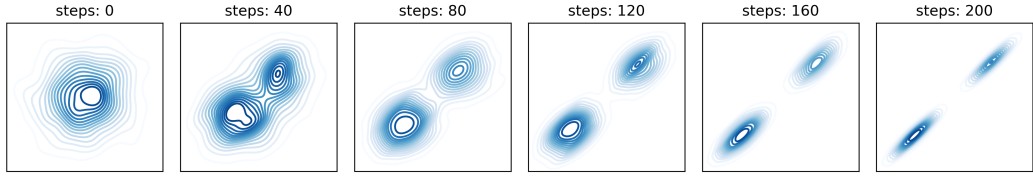

Figure 3: Diffusion *Reverse* process for 2 Gaussians.

## 4 CONCLUSION

GANs may suffer from mode collapse, resulting in less diverse samples, while DDPMs are capable of producing more diverse samples. Consequently, if sample diversity is critical, DDPMs may be more suitable than GANs. Noting that various types of GANs have attempted to address the problem of mode collapse. However, in practical settings, these approaches have limitations and have not fully resolved the issue. Nevertheless, there are obstacles to training and employing DDPMs, such as longer sampling times, which can be a limitation in certain applications. In the upcoming phases of this study, we plan to evaluate a newly developed Flow Matching network Lipman et al. (2022) that ensures stable training of continuous Normalizing Flows Chen et al. (2018).

URM STATEMENT

The authors acknowledge that at least one key author of this work meets the URM criteria of ICLR 2023 Tiny Papers Track.

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

## A EXPERIMENTS

### A.1 SYNTHETIC DATASET

This section presents the more detailed results of the experiments conducted on the synthetic dataset. These experiments were conducted to supplement the main experiments and provide a more comprehensive analysis of the performance of models .

### A.2 TWO GAUSSIANS

In this context, the modes of the distributions have samples of varying sizes. As illustrated in Figure 4, DDPM has the ability to generate samples from both modes, giving it an advantage over GAN.

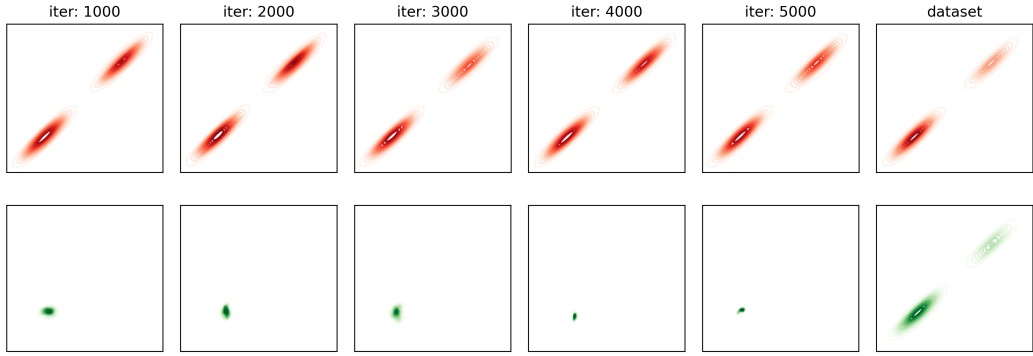

Figure 4: Training process of GAN (Green) and DDPM (Red) on a mixture of 2 Gaussians. GAN fails to make samples from the Gaussian with lesser samples, meanwhile DDPM keeps a relative balance between generated samples for each Gaussian.

### A.3 GRID GAUSSIANS

To further compare the methods, we did the experiment this time over a set of 9 Gaussians distributed over 3 by 3 grid with each Gaussian including 768 samples of data. The results, as demonstrated in Figure 5 show that GAN completely fails to replicate the original dataset, meanwhile, DDPM is capable of copying the grid distribution.

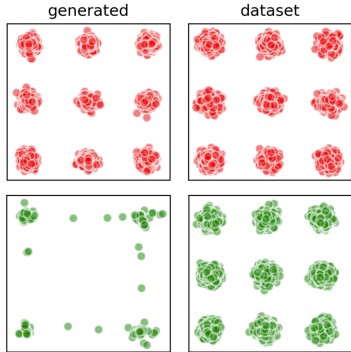

Figure 5: Generated samples for grid of Guassians. DDPM (red) successfully reproduces its original dataset (right column). Meanwhile, GAN (green) is missing the mode in center of the grid and has failed to preserve the uniform distribution of the grid dataset among the modes.

### A.4 DIAGONALLY DIRECTED GAUSSIAN

Furthermore, in order to evaluate the methods' capacity in reproducing the exact shape of the original datasets we used them to reproduce a diagonally directed Gaussian. As for this experiment the

Gaussian includes 2048 datapoints and the same number of samples are generated by each method. Once again as shown in Figure 6, DDPM outperforms GAN in regenerating the original dataset as one can see that GAN is more focused on reproducing certain parts is totally missing out on other sections of it. Please look at Appendix A.7 for experimental setups.

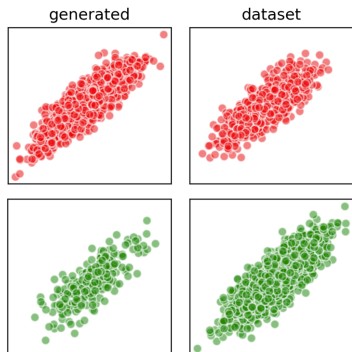

Figure 6: Generated samples for one diagonally directed Gaussian. DDPM (red) successfully reproduces the shape of the dataset (right column). Meanwhile, GAN (green) is unable to fully capture the range of the distribution.

## A.5 *Reverse* DIFFUSION PROCESS

In section, the dynamic of the *Reverse* diffusion process for two of synthetic datasets (8 and 2 Gaussians) and MNIST dataset are presented to show how these models work. The *Reverse* process of DDPM starts with pure noise and gradually moves the noise towards the true data distribution. Figure 7 shows multiple steps of this process for 8 symmetrically arranged Gaussians (top plot) and 2 disproportionately sampled Guassians (bottom plot). And Figure 8 shows this dynamic for the MNIST dataset.

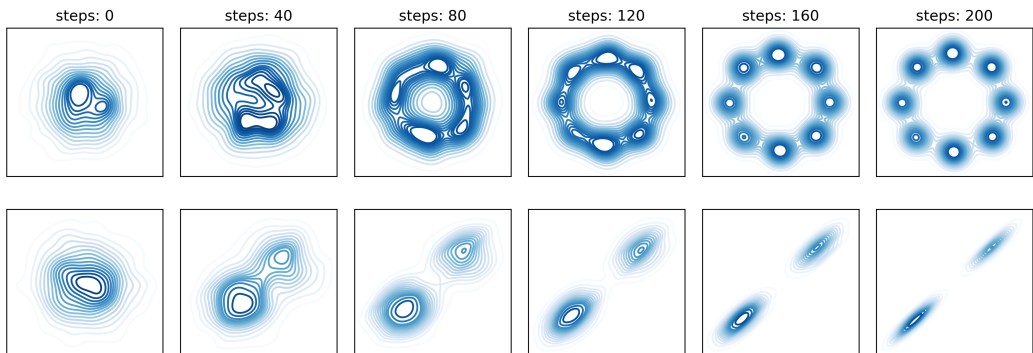

Figure 7: Diffusion *Reverse* process for 8 (top) and 2 (bottom) Gaussians datasets: the reverse process starts from noise and adds $T$ steps of denoising steps to generate samples that look like a true distribution.

## A.6 CHALLENGES

The effectiveness of a Denoising Diffusion Probabilistic Model (DDPM) for generating synthetic data depends on the initial random noise used for sampling. In order to capture every mode in the distribution, it is important to sample the noise from a range that covers the boundaries of the distribution in the synthetic datasets. For image data, it is typically recommended to normalize the input between [-1, 1] and initialize the noise from a Gaussian distribution that covers this range. This will ensure that the DDPM is able to cover the whole data distribution effectively. In comparison to a Valina Generative Adversarial Network (GAN), our experiments have shown that DDPM is

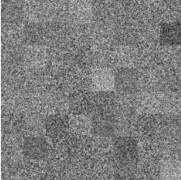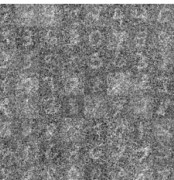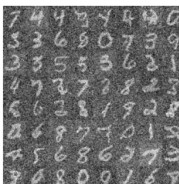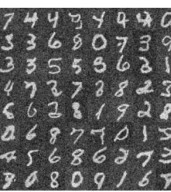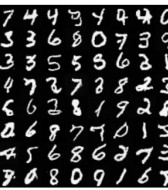

Figure 8: Diffusion *Reverse* process for the NNIST dataet: the reverse process starts from noise and adds $T$ steps of denoising steps to generate samples that look like a true distribution

generally able to cover the whole data distribution more effectively. However, it is important to be careful when selecting the initial random noise for sampling in order to achieve the best results.

Sampling from a DDPM involves applying $T$ denoising steps sequentially, which can take longer than sampling from a GAN. This is because sampling from a GAN only involves one pass through the generator network, whereas sampling from a DDPM involves multiple denoising steps, which can be computationally expensive. Therefore, if we want to generate a large number of samples in a short amount of time, it may be more efficient to use a GAN rather than a DDPM. This is particularly relevant for tasks such as generating high resolution images, where the computational cost of generating each sample can be high.

### A.7 EXPERIMENTAL SETUP

**Synthetic datasets:** For both models, we conducted experiments on synthetic datasets using an Adam optimizer Kingma & Ba (2014) with a learning rate of $1e{-}4$ and a batch size of $64$. We trained the models for 5000 epochs on each dataset.

**MNIST datasets:** To train the GAN we updated the network for 200 epochs with Adam optimizer Kingma & Ba (2014) with learning rate of $2e{-}4$ and batch size of $64$. And for DDPM, we trained the model for 100 epochs with Adam optimizer with learning rate of $0.001$ and batch size of $128$.

### B CODES

**DDPM:** We used and adapted the Diffusion Models repository by Cristian Garcia for synthetic datasets, and for the MNIST dataset we used and adapted the DDPM repository by Brian Pulfer.

**GAN:** We used the implementation of Vanilla GAN as suggested in the repo PyTorch-GAN by Erik Linder-Norén, and then adjusted the code so that it matches the implementations suggested in the Metz et al. (2016) paper for both synthetic and real settings.

## C  Background and Settings

### C.1  What is GAN?

A Generative Adversarial Network (GAN) is a type of deep learning model that is used for generating new, synthetic data that is similar to a given training dataset. While most generative models use maximum log-likelihood (or a lower bound), Generative Adversarial Models take a different approach by solving a min-max game Metz et al. (2016); Goodfellow et al. (2014). The adversarial framework consists of two multilayered networks: a *generative model G* and a *discriminator model D*. The generator network is trained to generate new data samples that are similar to the training dataset, while the discriminator network is trained to distinguish between real data samples from the training dataset and synthetic data samples generated by the generator.

The two networks are trained simultaneously (in an iterative manner that each takes one single step of optimization per time) in an adversarial process, with the generator trying to produce data that can fool the discriminator, and the discriminator tries to correctly identify whether a data sample is real or synthetic. Ideally, as the training process continues, the generator tends to make more realistic (similar to the real data space distribution) data. In the answer space for this min-max optimization there exist an equilibrium in which the generator is recovering all the real dataset and the discriminator's output is always 0.5 Goodfellow et al. (2014). Figure 9 shows the scheme of Vanilla GAN.

GANs have been used to generate a wide range of synthetic data, including images, audio, and text. They have been proven to be very effective at generating high-quality synthetic data and have been widely used in a variety of applications.

Defining the prior $p_z(z)$ over input noise, $G$ makes a mapping between noise space and the distribution $p_g$ where $D(x)$ represents the probability of $x$ belonging to the real-data distribution $p_{data}$ rather than $p_g$. That being said, the loss function is defined as belowGoodfellow et al. (2014).

$$\min_G \max_D \mathbb{E}_{x \sim p_{data}}[\log D(x)] + \mathbb{E}_{z \sim p_z}[\log(1 - D(G(Z)))]$$

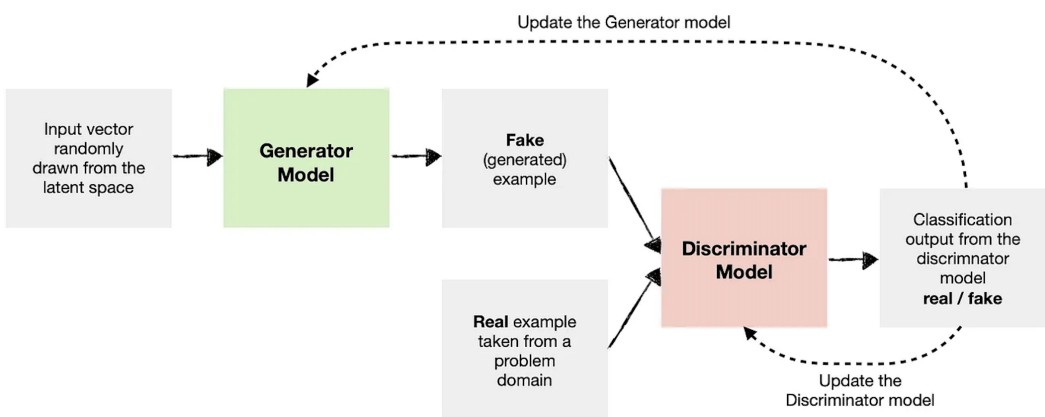

Figure 9: A simple scheme of Vanilla GAN. By taking as input the random noise in the latent space, the generator produces a fake sample in the space of real data. Then both synthetic and real data are fed into the discriminator. The loss of discriminator is determined by how well it detects the real data from the synthetic, the loss of generator by how good it is at deceiving the discriminator Dobilas (2022).

### C.2 WHAT IS DIFFUSION MODEL?

Diffusion models are latent variable models of the form $p_\theta(\mathbf{x}_0) := \int p_\theta(\mathbf{x}_{0:T}) \, d\mathbf{x}_{1:T}$, where each $\mathbf{x}_1, \dots, \mathbf{x}_T$ are latents of the same dimensionality as the data $\mathbf{x}_0$ Ho et al. (2020). This framework consists of two processes, *Forward* and *Reverse* diffusion processes. *Forward* diffusion process is fixed to a Markov chain that gradually adds Gaussian noise to the data according to a variance schedule $\beta_1, \dots \beta_T$, and *Reverse* diffusion process is the reverse process of adding noise to get a sample from data distribution starting with pure noise. The whole structure of the framework is illustrated In Figure 10, and these processes are defined as following.

**Forward process:** The goal of the forward process is to transform the original data into a noisy, distorted version of itself, which is then used as input for the reverse process. In the forward process of a Denoising Diffusion Probabilistic Model (DDPM), data is transformed through a series of diffusions in $T$ steps, which are random noise additions that are controlled by a variance schedule $\beta_1, \dots, \beta_T$.

The following process shows an example of addition Gaussian noise at every time steps, where the new example at timestep $t$ is defined by a Gaussian distribution centered at $\mu = \sqrt{1 - \beta_t} \mathbf{x}_{t-1}$ by variance of $\Sigma = \beta_t \mathbf{I}$ Weng (2021).

$$q(\mathbf{x}_t | \mathbf{x}_{t-1}) = \mathcal{N}(\mathbf{x}_t; \sqrt{1 - \beta_t} \mathbf{x}_{t-1}, \beta_t \mathbf{I})$$

$$q(\mathbf{x}_{1:T} | \mathbf{x}_0) = \prod_{t=1}^{T} q(\mathbf{x}_t | \mathbf{x}_{t-1})$$

**Reverse process:** The reverse process of a DDPM involves learning to reconstruct the original data from the noisy version produced by the forward process. This is typically done by applying a series of reverse diffusions, which are designed to undo the effects of the forward diffusions and reconstruct the original data.

A parameterized neural network tries to estimate the amount of noise added to a sample at timestep $t$ which will be used to denoise the sample.

$$p_\theta(\mathbf{x}_{0:T}) = p(\mathbf{x}_T) \prod_{t=1}^{T} p_\theta(\mathbf{x}_{t-1} | \mathbf{x}_t)$$

$$p_\theta(\mathbf{x}_{t-1} | \mathbf{x}_t) = \mathcal{N}(\mathbf{x}_{t-1}; \boldsymbol{\mu}_\theta(\mathbf{x}_t, t), \boldsymbol{\Sigma}_\theta(\mathbf{x}_t, t))$$

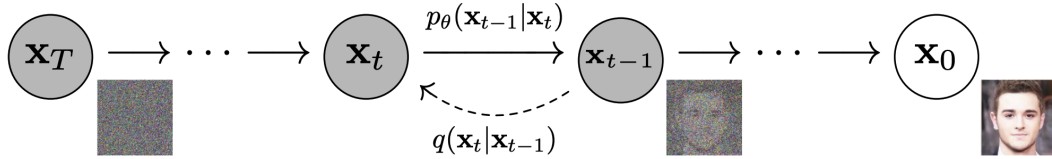

Figure 10: The Markov chain of forward and reverse diffusion processes of diffuion model Ho et al. (2020). In the *Forward* process the input $\mathbf{x}_0$ converts to random noise, and in the *Reverse* process the noise will be returned back to the data distribution through series of reverse diffusion steps.

After using variational lower bound, the following objective function is derived for training the model, where $\mathbf{x}_0$ is the input (e.g. image), $\boldsymbol{\epsilon}_t$ is the amount of noise added to the input at timestep $t$, and $\boldsymbol{\epsilon}_\theta(\mathbf{x}_t, t)$ is the predicted noise by the neural network at time step $t$.

$$L_t = \mathbb{E}_{t \sim [1,T], \mathbf{x}_0, \boldsymbol{\epsilon}_t} \left[ \| \boldsymbol{\epsilon}_t - \boldsymbol{\epsilon}_\theta(\mathbf{x}_t, t) \|^2 \right]$$

$$= \mathbb{E}_{t \sim [1,T], \mathbf{x}_0, \boldsymbol{\epsilon}_t} \left[ \| \boldsymbol{\epsilon}_t - \boldsymbol{\epsilon}_\theta(\sqrt{\bar{\alpha}_t} \mathbf{x}_0 + \sqrt{1 - \bar{\alpha}_t} \boldsymbol{\epsilon}_t, t) \|^2 \right]$$

For more details on the derivation of this objective function please look into Luo (2022).

The training and sampling algorithms of DDPM are shown in Algorithm 1 and 2 respectively.

---

**Algorithm 1** Training

---

1: **repeat**
2:   $\mathbf{x}_0 \sim q(\mathbf{x}_0)$
3:   $t \sim \text{Uniform}(\{1, \ldots, T\})$
4:   $\boldsymbol{\epsilon} \sim \mathcal{N}(\mathbf{0}, \mathbf{I})$
5:   Take gradient descent step on
      $\nabla_\theta \left\| \boldsymbol{\epsilon} - \boldsymbol{\epsilon}_\theta(\sqrt{\bar{\alpha}_t}\mathbf{x}_0 + \sqrt{1 - \bar{\alpha}_t}\boldsymbol{\epsilon}, t) \right\|^2$
6: **until** converged

---

Figure 11: **Training procedure of DDPM:** the process begins by selecting a sample from the distribution. Then, a random timestep is chosen and noise is added to the sample at that timestep. The model is then trained to remove this added noise from the corrupted sample.

---

**Algorithm 2** Sampling

---

1: $\mathbf{x}_T \sim \mathcal{N}(\mathbf{0}, \mathbf{I})$
2: **for** $t = T, \ldots, 1$ **do**
3:   $\mathbf{z} \sim \mathcal{N}(\mathbf{0}, \mathbf{I})$ if $t > 1$, else $\mathbf{z} = \mathbf{0}$
4:   $\mathbf{x}_{t-1} = \frac{1}{\sqrt{\alpha_t}} \left( \mathbf{x}_t - \frac{1-\alpha_t}{\sqrt{1-\bar{\alpha}_t}} \boldsymbol{\epsilon}_\theta(\mathbf{x}_t, t) \right) + \sigma_t \mathbf{z}$
5: **end for**
6: **return** $\mathbf{x}_0$

---

Figure 12: **Sampling procedure of DDPM:** a sample of pure noise is selected from a Gaussian distribution. Then, a series of denoising steps, numbered $T$, are applied to the sample. The result of these steps is a sample that is close to the true distribution.

## C.3   Scores

There is another point of view of the diffusion model which is the score-matching network. The idea is to parametrize the distribution as an energy-based model and use a score-matching network to learn the corresponding scores of the distribution which will be used for generating samples by Langevin dynamics.

The distribution is parametrized as the following format which is known as the Energy-based model, where $Z_\theta > 0$ is a normalizing constant, such that $\int p_\theta(\mathbf{x})d\mathbf{x} = 1$ Song (2021).

$$p_\theta(\mathbf{x}) = \frac{e^{-f_\theta(\mathbf{x})}}{Z_\theta}$$

$$Z_\theta = \int p_\theta(\mathbf{x})d\mathbf{x} = \int e^{-f_\theta(\mathbf{x})}d\mathbf{x} = 1$$

There are several ways to train energy-based models: maximum likelihood training with Markov Chain Monte-Carlo (MCMC), Score Matching, Noise Contrastive Estimation, etc. A detailed overview of these training methods can be found in Song & Kingma (2021). But the denoising score matching showed promising results in many applications such as image generation Song et al. (2020).

Evaluating the energy-based model is hard due to difficulty of calculating the normalizing constant $Z_\theta$; we can omit to evaluate it using score-based model and learn the scores of the distribution instead.

$$\mathbf{s}_\theta(\mathbf{x}) = \nabla_\mathbf{x} \log p_\theta(\mathbf{x}) = -\nabla_\mathbf{x} f_\theta(\mathbf{x}) - \underbrace{\nabla_\mathbf{x} \log Z_\theta}_{=0}$$

$$= -\nabla_\mathbf{x} f_\theta(\mathbf{x})$$

Then we can generate new samples by using learned scores along with the Langevin dynamic as follows:

$$\mathbf{x}_{i+1} \leftarrow \mathbf{x}_i + \epsilon \nabla_\mathbf{x} \log p(\mathbf{x}) + \sqrt{2\epsilon}\, \mathbf{z}_i, \quad i = 0, 1, \cdots, K$$

This dynamic is pretty similar to what we had before in DDPM, where the score function $\nabla_{\mathbf{x}} \log p_\theta(\mathbf{x})$ can be considered as the denoising model, $\boldsymbol{\epsilon}_\theta(\mathbf{x}_t, t) = \nabla_{\mathbf{x}} \log p_\theta(\mathbf{x})$.

In order to show the dynamic of score-based generative model, we have trained a score-based model on **Swiss roll** dataset. Figure 13 shows the swiss roll dataset (left), generated sample (middle), and scores (right). As it is clear in the figure, scores point towards the high-density regions of the distribution; accordingly, the Langevin dynamic moves every sample in the space towards the distribution and eventually generates samples close to the true distribution.

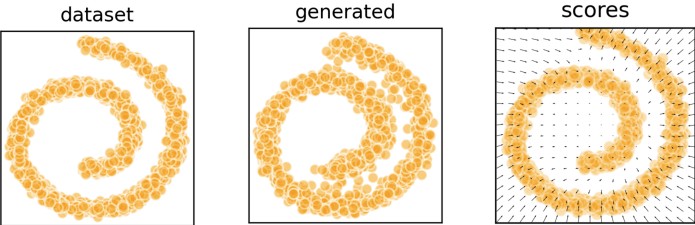

Figure 13: An example of a score-based model for modeling a 2D swiss roll data: dataset (left), generated sample (middle), scores (right). For generating samples, multiple uniform random noises are generated between the boundaries of the dataset (maximum distance between two points in the dataset), and then by applying $T$ (e.g. 200) steps of Langevin dynamics sample moved toward the high density of the distribution.

### C.4 WHAT IS MODE COLLAPSE?

A well-trained GAN can generate a well-spread sample space in real data dimensions. However, a very recurring problem while training a GAN is to see the generator making a small subset of the distribution. For a generator producing the same output, the best strategy for the discriminator is to reject that output. However, for a discriminator stuck in a local minimum, the generator's best strategy is to find the point most plausible for the discriminator to detect as real data. So this effect causes the generator to over-optimize on a single point at each step, and this results in the generator switching between small sets of distribution. This observation is defined as *mode collapse* mod (2022). Variants of GAN have been proposed in order to tackle the instability in training of GANs, naming UnrolledGAN Metz et al. (2016), and Wasserstein GAN (WGAN) Arjovsky et al. (2017). Briefly explaining, UrolledGAN attempts to solve the issue of generators over-optimization by incorporating current discriminators output along with multiple updates of discriminator. This way generator is taking advantage of future discriminators instead of over-optimizing on the current discriminator. Also, in WGAN this issue is solved through adjusting the loss function for the discriminator [1] such that the vanishing gradient problem won't happen. In other words a more agile (not stuck in local optima) discriminator would be able to stop the generator from over-optimization on a single data point in the original dataset, hence, avoid the mode collapse.

---

[1] The discriminator in WGAN doesn't output a number between 0 and 1. Instead it only tries to make the output bigger for real samples than generated ones. Hence it is in fact called "critic" Arjovsky et al. (2017).

