# OpenReview forum: "A Study on Sample Diversity in Generative Models: GANs vs. Diffusion Models"
_ICLR.cc/2023/TinyPapers — Submitted to Tiny Papers @ ICLR 2023_

### Official Review · Reviewer_PMxY · 2023-03-22

**Confidence:** 4

**Summary Of Contributions:**

This paper is an empirical study focusing on the sample diversity of two representative generative models (i.e., GAN and DDPM), and the connection between it and mode collapse. Two sets of experiments are conducted on tiny datasets to demonstrate that DDPM is more powerful to generate diverse samples than GAN.

**Rating:**

Clear, Correct, and Reproducible (CCR): a submission which meets the reviewing criteria

**Strengths And Weaknesses:**

**Strengths:**

* This paper provides comprehensive preliminaries about generative models in the appendix.
* The paper is generally clear and well-structured.
* Experimental findings justify the conclusion.

**Weakness:**

* Most of the images are blurry when zooming in. The authors should use pdf instead of (high-resolution) png/jpg.
* The authors should clearly state which color is the result of GAN/DDPM in Figures 1, 2, and 4.
* The current experiments are only focusing on observing the results/phenomenon. Further (deeper) analyses are needed. For example, why DDPM is better than GAN in terms of sample diversity? *Either empirical or theoretical insights could significantly improve the paper.*

**Minor:**

* In Appendix C.1,

  * "They have proven to be very effective ..." -> "They have been proven".

  * The author is suggested to improve the writing, e.g., avoid writing long sentences.

    > "The generator outputs a fake sample in real data space generated by taking in random noise in its latent space." in the caption of Figure 9. -> "By taking as input the random noise in the latent space, the generator produces a fake sample in the space of real data."

* In Appendix C.2,

  * "Reverse diffusion process... **form** ..." -> from.
  * "which will be **sued** to denoise the sample" -> used.
  * "..., **which** x0 is the input" -> where.

**Overall:**

* *Clarity:* Fair. The paper is clearly written. However, I suggest adding more discussions of relevant literature.

* *Correctness:* Good. Most of the paper is correct. But I didn't check the correctness of Appendix C.3, in which I am not an expert.

* *Reproducibility:* Good. The author doesn't provide the source code. But I think the results are reproducible based on the presented setups.

* *Follows basic requirements:* Yes, the authors follow the ICLR code of conduct.

**Suggested Changes:**

Please see the weakness and minor sections. About the writing, I suggest the authors trying with AI-assisted rewriting tools, such as Grammarly and ChatGPT.

---

> ### Author Response · Authors · 2023-05-23
> **Addressing Feedback**
>
> Dear Reviewer,
>
> Thank you for your valuable feedback on our paper. We appreciate your thorough evaluation and suggestions. We have considered your comments and made the necessary improvements to enhance the quality of our work, including addressing the concerns raised, improving image clarity, and making the minor changes you mentioned.
>
> In light of your comment about providing deeper analyses on why DDPM outperforms GAN in terms of sample diversity, we kindly request your suggestions on specific approaches or experiments that could contribute to a more comprehensive understanding of this comparison.
>
> We would like to express our gratitude once again for your valuable input. We believe that the revised version of our paper incorporates these improvements. If you have any further questions or require additional information, please feel free to reach out to us.
>
> Sincerely,

---

### Official Review · Reviewer_PiXY · 2023-04-03

**Confidence:** 4

**Summary Of Contributions:**

The authors study the sample diversity in GAN vs. diffusion models by conducting experiments using both synthetic and image data to explore the connection between mode collapse and sample diversity in GAN and diffusion models.

**Rating:**

Great Start (GS): a submission which meets some of the reviewing criteria but has room for improvement

**Strengths And Weaknesses:**

Strength:
1. Paper is clearly written. Motivation is also quite clear.

Weakness:
1. Experiments were carried out on simplistic synthetic data or simplistic image data such as MNIST. In order to draw meaningful conclusions, especially with respect to sample diversity, more complex datasets need to be explored, such as ImageNet.


**Suggested Changes:**

Add Imagenet experiments.

---

> ### Author Response · Authors · 2023-05-21
> **Exploring the Intuitive: Leveraging Simplistic Datasets for Clearer Insights**
>
> Dear reviewer,
>
> Thank you for your review and valuable feedback on our paper. We appreciate your positive comments regarding the clarity of the writing and the clear motivation behind our work.
>
> Regarding your concern about the simplicity of the datasets used in our experiments, we understand your point. We chose to use simpler datasets, such as Synthetic points and MNIST, primarily for their intuitiveness and ease of understanding. These datasets are widely used in the research community as a starting point to demonstrate and validate novel ideas due to their simplicity and well-established benchmarks.
>
> However, we acknowledge that in order to draw more comprehensive conclusions, it would indeed be beneficial to explore more complex datasets, such as ImageNet.
>
> Unfortunately, due to limited resources and computational constraints, training models on datasets like ImageNet can be highly demanding. As a result, we decided to focus on the simpler datasets to provide a clear and intuitive understanding of our studies. Nevertheless, we recognize the importance of expanding our experiments to more complex datasets in future research to obtain a more comprehensive evaluation.
>
> Once again, we appreciate your thoughtful feedback, and we will consider your suggestions for further improvement in our future work.
>
> Sincerely,

---

### Comment · Area_Chair_a8oY · 2023-06-06
**Check for Archival**

This work meets the threshold for archival, contents the URM statement and is deanonymized.

---

### Meta-Review · Area_Chair_a8oY · 2023-04-02

**Recommendation:** Invite to present
**Confidence:** 3

**Metareview:**

This paper studies the sample diversity of generative models and makes connections between mode collapse and sample diversity. The authors empirically demonstrate that DDPMs have a better sample diversity than GANs, and could preserve a comprehensive representation of the data distribution.

This paper only receives one review (AC will adjust the meta-review If another review comes in later). The reviewer acknowledges that the paper meets the CCR standard. Some minor issues exist, such as blurry images and confusing captions/legends. Please fix them in the final version. Moreover, deeper analyses and insights could further improve the paper.

Overall, based on the review criteria of the ICLR TinyPaper Track, it meets the CCR standard. However, the writing of this paper could be further polished. Please carefully revise and proofread the paper following the reviewer's comments.

----

Update: We have received another review. The reviewer's concern is about the tiny experimental validation. Therefore, please conduct extra experiments on more complex datasets such as ImageNet. My final recommendation remains the same.



**Summary:**

This paper studies the sample diversity of generative models and makes connections between mode collapse and sample diversity. The authors empirically demonstrate that DDPMs have a better sample diversity than GANs, and could preserve a comprehensive representation of the data distribution.

**Comments And Feedback To The Authors:**

Please carefully revise and proofread the paper following the reviewers' comments.

**Reason For Not Giving A Higher Recommendation:**

* Lack of deeper insights.

**Reason For Not Giving A Lower Recommendation:**

* This paper meets the CCR standard. Most of the content is clearly written.

* The claims are well justified.

---

### Decision · Program_Chairs · 2023-04-07

Invite to present